# The influence of language proficiency, acculturation stress, and institutional support in enhancing personal development of international students in China

**Muhammad Azram** [1]*, **Mei Hong**[1], **Waqar Ahmad**[2], **Ali Sohail**[1], **Bilal Ahmad**[3]

**1** School of Public Policy and Administration, Xi'an Jiaotong University, Xi'an, China, **2** Faculty of Management Sciences, University of Central Punjab, Lahore, Pakistan, **3** Department of Mathematics and Statistics, Hazara University, Mansehra, Pakistan

* azram_khan@stu.xjtu.edu.cn

## Abstract

This study highlights the significance of understanding how diversity experiences and innovative learning environments contribute to the personal development of international students in Higher Education Institutions (HEIs) in Shaanxi, China. Despite existing studies, there remains a notable gap in exploring their combined effects specifically within Chinese HEIs. This research aims to address this gap by analyzing the relationships among diversity experiences, language proficiency, and personal development, thereby offering valuable insights for educators and policymakers to enhance institutional support for international students. To achieve this, we conducted a quantitative survey involving 364 international students, utilizing a structured questionnaire and employing Structural Equation Modeling (SEM) for data analysis. The results reveal a positive relationship between diversity experiences and personal development. However, no direct relationship was found between innovative learning environments and personal development. Furthermore, language proficiency mediates the relationship between diversity experiences and personal development. Moreover, innovative learning environments were positively associated with both acculturation stress and language proficiency, which in turn were related to personal development. The findings underscore the vital role of institutional support in enhancing international students' personal development through improved diversity experiences and language proficiency. To achieve this, educators and policymakers should integrate diversity initiatives and create adaptable learning environments that specifically address the needs of international students.

## 1. Introduction

China has gradually positioned itself as a projecting worldwide education center, fascinating a significant number of international students to its HEIs. With over 492,000 international students registered as of recent decade, China has developed one of the leading spot for higher education worldwide [1]. This incursion of foreign students presents both potential and

**Data availability statement:** All relevant data are uploaded in supporting material.

**Funding:** The author(s) received funding from **Major Project of the National Social Science Fund of China**: "Research on Key and Challenging Issues in Coordinated Promotion of the Integrated Reform of the Education, Science and Technology, and Talent Systems and Mechanisms" (Project Number: 24ZDA028) and **National Natural Science Foundation of China Project**: "Research on the Diversity of Scientific Research Teams and Its Impact on Innovative Leadership from the Perspective of Resource Integration" (Project Number: 72372126).

**Competing interests:** No authors have competing interests.

limitation for the international students and the academies that host them. One of the top priorities is the personal development of these learners, specifically as they adjust to new cultural, social, and academic environments. Personal development raises to the comprehensive development of individuals in terms of intellectual, emotional, and social capacities, which is critical to their academic accomplishment and overall comfort [2].

A crucial component shaping international students' personal development is their diversity experiences, which include connections with colleagues from diverse cultural, ethnic, and linguistic contexts. Such practices are identified to improve intercultural skills, critical thinking, and adjustability, skills that are vigorous for achievement in expanding globalized educational and professional settings [3]. Though, experience to diversity can also prime to obstacles, mainly when learners encounter language fences, cultural miscaptions, or social detachment due to a deficiency of institutional assistance. Without abundant resources, foreign students may fight to deeply involve with their academic and social landscapes, restraining the potential reimbursements of their diverse connections. Innovative learning environments are one more noteworthy aspect conducive to personal development. These atmospheres are considered by the use of technology-infused education, group learning framework, and student focused approaches that inspire creativity and critical thinking [4]. Whereas such atmospheres promote deeper engagement and educational accomplishment, adjusting to new learning methods and technologies can be daunting for foreign students, mostly those who come from schooling system that may not value these tactics [1]. This highlights the crucial for educational institutions to distribute specialized assistance to students adjusting and maximizing innovative learning opportunities.

Language proficiency play a significant role in building foreign students' social and learning experiences. Mastery of the host country's language is crucial for effective communication, active participation in learning, and building impactful relationships. Research shows that foreign students fronting challenges with language barriers are likely to experience acculturation stress, a form of psychological pressure that rises when people struggle to familiarizing to a new cultural setting [5]. Significant levels of acculturation stress can delay academic performance and influence personal development, highlighting the need for organizations to provide comprehensive language support and integration programs that affluence the cultural adaptation phase [6]. Under these circumstance, institutional support grows a critical factor in illustrating the difficulties inherent in diversity experiences, innovative learning environments, and language proficiency. Effective institutional support involves a range of services, including academic guidance, psychological support, language proficiency, and cultural immersion initiatives. Learning institution that provide sustainable support services, that permit foreign leaner to steer cultural and learning shifts more effectively, easing acculturation stress and nurturing greater personal development [7]. By facing these hurdles, organizations can improve international students' learning outcomes and overall well-being.

Despite research on foreign students' experiences has full-fledged in recent decade, there are still gaps in thoughtful how influences such as diversity experiences, innovative learning environments, language proficiency, acculturation stress, and institutional support cooperate to stimulus personal development, mainly in the scope of Chinese HEIs. Many studies review these aspects autonomously, without seeing the intricate links between them. This inquiry aims to discourse this gap by sightseeing how these variables communally form the personal development of international students in China, offer deeper understanding of their educational and social interactions.

The conclusions from this study will have substantial implications for educators, administrators, and policymakers. By recognizing the key aspects that stimulus international students' personal development, this research will clarify the launch of targeted support scheme that

improve educational success and facilitate smoother cultural adaptation. Organizations can grow integrated services that comprise academic and mental health support, language counselling, and inclusive pedagogical approaches. Policymakers can use these understandings to form national strategies that indorse the integration and success of international students, contributing to more effective and inclusive academic settings.

## 2. Literature review

### 2.1 Diversity experiences

Diversity experiences, explained as connections with colleagues from different cultural, ethnic, and linguistic roots, are usually known for their enriching contributions to foreign students' personal development. Leask [8] draws attention to such experiences foster cross cultural skills, critical thinking, and flexibility, which are important in international education and career landscapes. Immersion to diverse perspectives prompts students to interact with different global outlook, expanding their capacity to problem-solve and versatile approach to challenges. Though, diversity experiences also bring forth challenges. Tavares [9] indicate that cultural differences can occasionally can cause social detachment and misunderstandings, expressly when foreign students face difficulties to integrate with local peers. This risk of exclusion is exacerbated when organizations do not foster adequate assistance to facilitate cross cultural connections. For resource-limited students, diverse environments may become a source of stress rather than development.

### 2.2 Innovative learning environments

Innovative learning environments are designed to encourage active learning, creativity, and collaboration. These environments, categorized by the integration of technology and student-centered pedagogies, support students advance critical thinking and problem-solving skills, enhancing both success in academic and fostering personal development [10]. Particularly, technology-supported learning environments offer occasions for students to involve deeply with course content through interactive and collaborative tools. However, Heng [11] claims that foreign students, mostly those from academic systems that highlight traditional schooling methods, may brawl to adapt to these new environments. Students who are not aware with digital educational tools or interactive teaching practices may experience stress and lack of engagement, eventually limiting the benefits of these innovative approaches. Organizations that flop to backing such students risk spreading the gap between students' academic backgrounds and the demands of advance educational environments.

### 2.3 Language proficiency

Language proficiency has a key role in foreign students' educational integration and social success. Advance language skills permits students to participate actively in classroom deliberations, engage with academic content, and build social network, which cooperatively decrease acculturation stress [12]. Language proficiency also improves students' confidence, allowing them to link more efficiently and stunned cultural fences. Though, Yan and Berliner [13] emphasize that students with low language proficiency face significant challenges. These students are more likely to experience academic struggles, social isolation, and acculturation stress, as they may feel disconnected from their peers and unable to fully engage with academic materials. Language barriers not only affect students' academic performance but also limit their ability to form supportive social networks, which are crucial for coping with the challenges of studying abroad.

## 2.4  Acculturation stress

Acculturation stress arises when foreign students face challenges in adjusting to the cultural standards and customs of the host country. Smith and Khawaja [12] identify acculturation pressure as a mutual issue for foreign students, mainly during the beginning of their transition. Students facing advance levels of acculturation stress could struggle with anxiety, depression, and educational difficulties, which obstruct their personal development. From a positive side, Lashari et al. [14] propose that organizations that offer cultural induction programs, peer support system, and psychological support services can meaningfully mitigate the negative influence of acculturation stress. these institutional initiatives support students adjust to the new cultural environment, assisting the adjustment and stimulating a healthier and more productive educational experience. Regardless of this, many institutions do not have robust programs in place to meet the mental health requirements of foreign students.

## 2.5  Institutional support

Institutional support is a critical element affecting foreign students' achievement and personal development. Organizations that provide inclusive support services, plus educational counseling, behavioral training, and language coaching, are better able to provide an setting where foreign students can flourish [15]. These facilities not only assist students steer educational difficulties but also backing their societal and emotional well-being, conducive to their overall accomplishment. Notwithstanding the familiar standing of institutional support, numerous HEIs fall short of providing the essential possessions to highlights the various demands of their foreign student populations. Azram et al. [1] argue that while some educational institutions have vigorous support systems in place, others focus primarily on educational performance, overseeing the social and emotional difficulties that foreign students frequently face.

## 2.6  Hypotheses development

This section discusses the associations between diversity experiences, innovative learning environments, language proficiency, acculturation stress, and institutional support, with respect to their influence on the personal development of international students. Based on an exhaustive literature review, the following hypotheses are formulated.

## 2.7  Diversity experiences and personal development

Diversity experiences, identified as engagement with students from various cultural, ethnic, backgrounds, and linguistic are recognized for their positive impression on personal development [16]. These practices foster superior intercultural capability, adaptability, and rational thinking [17]. Exposure to diverse views inspires students to extend their global outlook and involve in introspection, which are crucial elements of personal growth [18]. Therefore, we hypothesize:

   H1**:** *Diversity experiences positively influence the personal development of international students.*

## 2.8  Innovative learning environments and personal development

Innovative learning environments characterized by active, student-centered tactics and integration of technology, have been found to improve innovative thinking, problem-solving, and reflective thinking [1,18]. These atmospheres motivate more profound engagement and

collaborative efforts, which nurture personal development. Though, adjusting to innovative learning environments can be stimulating, particularly for students from traditional academic systems [19]. Thus, we hypothesize:

H2: *Innovative learning environments positively influence the personal development of international students.*

## 2.9  Diversity experiences and language proficiency

Students who interact with diverse colleagues are further likely to improve their language proficiency, as they frequently practice and communicate in a second language [20]. Language acquisition is accelerated through social connections in cross-cultural environments, which encourage both formal and informal educational pathways [21]. Based on this, we hypothesize:

H3**:** *Diversity experiences positively influence the language proficiency of international students.*

## 2.10  Diversity experiences and acculturation stress

Diversity experiences offer students with enriching experiences to involve with colleague from diverse cultural backgrounds, which can support decrease acculturation stress by indorsing superior cultural awareness and flexibility [22]. These connections inspire students to grow multicultural linguistic competencies and widen their outlooks, making it relaxed for them to join in into their new setting and accomplish the encounters of cultural adaptation [23]. Thus, we hypothesize:

H4: *Diversity experiences positively influence acculturation stress among international students.*

## 2.11  Innovative learning environments and language proficiency

Innovative learning environments, which comprise technological platforms and cooperative strategies, deliver foreign students with abundant opportunities to advance their language proficiency [24]. By fetching in collaborative learning tasks and peer networking, students are gifted to training and improve their language competencies [25]. Therefore, we hypothesize:

H5**:** Innovative learning environments positively influence the language proficiency of international students.

## 2.12  Innovative learning environments and acculturation stress

For foreign students, adjusting to innovative learning environments can support helps alleviate acculturation stress by creating opportunities for engagement, collaboration, and communication with colleague and mentors [26,27] (Johnson et al., 2018; Wilczewski & Alon, 2023). Fostering learning settings that embrace collaborative technologies and student-oriented pedagogies inspire students to aggressively contribute in their learning, which can improve their confidence and decrease the stress related with cultural adoption [12]. So, we hypothesize:

H6**:** Innovative learning environments positively influence acculturation stress among international students.

## 2.13  Language proficiency and personal development

Language proficiency is vital for academic achievement and social inclusion, both of which are vital for personal development [28]. Students with advanced language competencies are important to contribute in class thoughts, form social influences, and steer their new cultural atmospheres, principal to improved personal development [12,29]. Therefore, we hypothesize:

H7: *Language proficiency positively influences the personal development of international students.*

## 2.14  Acculturation stress and personal development

Acculturation stress, which outcomes from the difficulties of familiarizing to a new culture [23], can negatively affect students' learning performance and personal development [30]. Strong presence of acculturation stress are related with anxiety, social separation, and trouble to adjusting to the host country culture, all of which delay personal development [31]. Therefore, we propose:

H8: Acculturation stress negatively influences the personal development of international students.

## 2.15  Mediating roles of language proficiency and acculturation stress

Students with unconventional language proficiency are improved equipped to involve in expressive connections with diverse colleague, which improves their personal development [32]. Hence, we hypothesize:

H9: *Language proficiency mediates the relationship between diversity experiences and personal development.*

Furthermore, language proficiency and innovative learning environments language proficiency is also predictable to mediate the association between innovative learning environments and personal development. Students with unconventional language skills can more efficiently contribute to the collaborative and technology-driven learning environments, which further contributes to their personal development [33]. Hence, we hypothesize:

H10**:** *Language proficiency mediates the relationship between innovative learning environments and personal development.*

Furthermore, acculturation stress as a mediator acculturation stress mediates the relationship between diversity experiences and personal development. When students experience strong presence of acculturation stress, the paybacks of diversity experiences may be lessened as stress constrains their capability to involve completely [34]. Henceforth, we hypothesize:

H11**:** *Acculturation stress mediates the relationship between diversity experiences and personal development.*

Besides, acculturation stress and innovative learning environments acculturation stress is also expected to mediate the connection between innovative learning environments and personal development. High levels of stress can prevent students from benefiting from innovative learning environments, reducing their potential for personal development [35]. Thus, we hypothesize:

 H12: Acculturation stress mediates the relationship between innovative learning environments and personal development.

### 2.16 Moderating role of institutional support

Institutional support, which includes learning counselling, language backing, and mental health services, is a crucial factor in enhancing students' ability to cope with acculturation stress and advance their language proficiency [36,37]. Institutions that deliver inclusive backing schemes produce an environment that nurtures personal growth and learning achievement [1]. Therefore, we hypothesize:

 H13: *Institutional support moderates the relationship between diversity experiences and personal development, enhancing the positive effects of diversity experiences.*

 H14: *Institutional support moderates the relationship between innovative learning environments and personal development, enhancing the positive effects of innovative learning environments.*

 H15: *Institutional support moderates the relationship between language proficiency and personal development, amplifying the benefits of higher language proficiency.*

 H16: *Institutional support moderates the relationship between acculturation stress and personal development, reducing the negative impact of acculturation stress.*

## 3. Theoretical framework and methodology

### 3.1 Theoretical framework

This investigation is based in the dynamic influence of social cognitive theory (SCT) [38] and ecological systems theory (EST) [39], delivering an extensive examination through which to sightsee the personal development of foreign students in HEIs in Shaanxi, China.

 SCT draw focus to the critical influence of observational learning, social engagement, and self-efficacy in determining individual performance and development. In this scenario, foreign students' experience to diversity experiences and innovative learning environments offers rich foundation for developing new skills, familiarizing to challenges, and educating self-confidence. These atmospheres consent students to learn by charming with colleague from diverse cultural and language experiences, nurturing cultural flexibility and personal development. Additionally, language competencies, as an essential component of self-efficacy, permits students to direct both learning and social settings with greater flexibility, while institutional support aids alleviate acculturation stress and makes an encouraging environment for students to flourish. By connecting personal agency with ecological influences, SCT delivers a robust agenda for sympathetic how students actively form and are shaped by their experiences.

 On the other hand, EST covers the opportunity by highlighting how diverse environmental systems cooperate to affect an individual's development. The essence of this study is the recognition that foreign students are not only formed by their instant learning and

social environments (the microsystem) but also by the connections between these situations (the mesosystem) and the broader provision networks provided by their organizations (the exosystem). For instance, institutional support holds key component in fostering students accomplish the adaption complexities to a new academic structure and social environment. The macrosystem, which incorporate societal norms, cultural expectations, and broader worldwide influences, further effects how students experience and answer to acculturation stress. By positioning international students within these interlinked structures, Ecological Systems Theory underlines the standing of a supportive network in nurturing their personal development.

By uniting SCT and EST, this learning offers an ironic, complex view of how personal and environmental factors touch the international student's personal development. Diversity experiences and innovative learning environments are not distinct influences but are firmly rooted in a vast network of social and institutional framework. This integrated dynamic not only show the significant connection between students' individual capacities, such as language competency, and their auxiliary support structures, but also offers a discriminating under-standing of how students shift, enhance, and succeed in diverse and evolving academic set-tings. In this context, it points out the transformative potential of robustly validated learning environments, where students are prepared to navigate difficulties and comprehend their full potential, as shown in Fig 1.

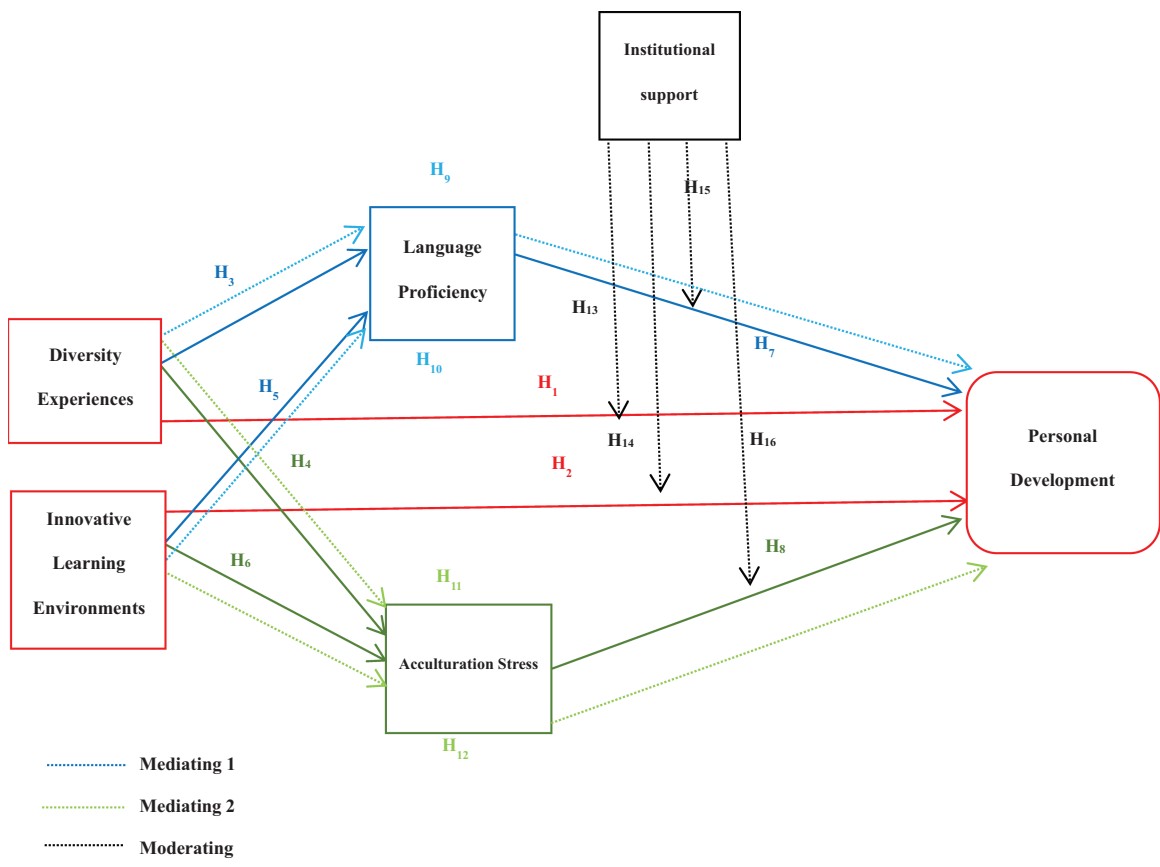

**Fig 1. Framework of the study.**

## 3.2  Research methodology

This study attentions on the mediating roles of language proficiency and acculturation stress, as well as the moderating role of institutional support in HEIs. The research examines the sophisticated interactions between diversity experiences, innovative learning environments, and students' personal development in Shaanxi, China. The research methodology is enlightened in detail below:

## 3.3  Research design

The study hires a quantitative study design, selected for its capability to rigorous gather and analyze numerical data. This configuration allows for a coherent examination of the relationships between diversity experiences, innovative learning environments, language proficiency, acculturation stress, and institutional support. The quantitative approach is ideally fit for exploring the elaborate interactions among these variables, as it allows the use of innovative statistical techniques, such as SEM.

## 3.4  Participants and sampling

The participants were recruited from a population of international students presently enrolled in HEIs in Shaanxi, China. This group was selected because of their diverse backgrounds and experiences, which are vital to recognizing the influence of diversity experiences and innovative learning environments on personal development. The choice of participants was built on the relevance of their experiences to the research questions, mainly regarding their adaptation to new cultural and educational settings.

Convenience sampling was utilized to select participants. This design was chosen due to its practicality in reaching foreign students from several organizations within Shaanxi. although convenience sampling present certain limitations regarding generalizability, it permitted the study to efficiently capture a broad spectrum of experiences from a diverse student body. A total of 364 participants took part in the study, which constitutes an appropriate sample size for SEM analysis. The sample size was determined based on guidelines for SEM, which requires large samples to ensure the reliability and validity of the findings.

## 3.5  Pilot study

Prior to distributing the complete survey, a trial study was executed with 30 participants, who inflected the target population well. The trial study intended to test the transparency, reliability, and validity of the questionnaire. The pilot participants feedback was used to make slight modification to the questionnaire. This approach ensured that the questionnaire was clear and effective in acquiring the necessary information, boosting the overall reliability of the results.

## 3.6  Survey instrument

The questionnaire employed in this research was adapted from established, validated scales from previous studies. The survey was divided into two sections: one for gathering demographic information and the other for measuring key constructs using a five-point Likert scale (1 = Strongly Disagree to 5 = Strongly Agree). The constructs were adapted from reputable sources, including diversity experiences [40,41], innovative learning environments [1], language proficiency [14], acculturation stress [42], institutional support [43] and personal development [44,45], established from relevant studies.

### 3.7 Data collection and analysis

Participants responded to the survey using the Likert scale to express their level of agreement with the showed statements. The data collected were analyzed using Smart PLS4, with SEM executed to inspect the associations between the variables. SEM was preferred because it allows for the evaluation of both direct and indirect effects between multiple variables and is mostly well-suited for testing intricate models like the one proposed in this study. To certify the rationality of the SEM analysis, several key assumptions were verified. Linearity between variables was checked using scatterplots, and multivariate normality was assessed through Mardia's test. While there were minor deviations from normality, SEM is vigorous to these deviations when sample sizes are large [46]. Multicollinearity was evaluated using the Variance Inflation Factor (VIF), which showed no significant collinearity issues. Model fit indices, including Chi-square, CFI, and RMSEA, indicated an acceptable model fit.

### 3.8 Ethical considerations

Ethical approval was obtained from the relevant institutional review boards of Xi'an Jiaotong University prior to the commencement of the study. Participants were provided with detailed information about the purpose and methods of the research and were required to give informed consent before participating. Confidentiality and anonymity were maintained throughout the research process, ensuring that participants' data were handled with the highest level of ethical integrity.

## 4. Results

This section presents the findings of the study and interprets their significance. The results are analyzed to understand the relationships between the key variables. Table 1 shows the factor loadings for each study variable, highlighting significant contributions across components. Furthermore, the demographic analysis of the study is carried out by employing 18 questions that are provided in supplementary materials.

The statistics in Table 2 suggest that the constructs measuring have high internal consistency (as indicated by Cronbach's α, $\rho_a$, and $\rho_c$) and that they explain a significant portion of the variance in their respective items (as indicated by AVE). This is generally a positive sign of the reliability and validity of the measurement instruments used in the study. It implies that the constructs are well-defined and can be trusted as reliable measures for the study.

Table 3 displays various fit indices obtained from a SEM investigation. These fit indices are essential for assessing how well the model aligns with data. Both models provide a good model fit, with the SRMR (Standardized Root Mean Residual) values lying well below the suggested threshold of 0.08. Both of the models' NFI (Normed Fit Index) values are above the suggested threshold of 0.9, suggesting a satisfactory fit. On the other hand, the first model appears to match the data slightly better, as indicated by its slightly higher NFI value.

In Table 4, the Fornell-Larcker criterion table shows the measurement model has good discriminant validity, as each construct is distinct and explains more variance in its measured variables than it shares with other constructs. This is an important finding in SEM as it ensures that the constructs are meaningfully different from each other.

In Table 5 the relationship between diversity experience and personal development is statistically significant (β = 0.267, p = 0.000). The beta coefficient (β) of 0.267 suggests that a 1-unit change in diversity experience is expected to bring about a 26.7% change in personal development, which supports hypothesis $H_1$. The relationship between innovative learning environment and personal development is not statistically significant (β = 0.098, p = 0.140), thus these results do not support hypothesis $H_2$. The relationship between diversity experience

**Table 1. Factor loading of study variables.**

| Variables | Acculturation Stress | Diversity Experience | Innovative Learning Environment | Institutional Support | Language Proficiency | Personal Development |
|---|---|---|---|---|---|---|
| AS 1 | 0.910 | | | | | |
| AS 2 | 0.887 | | | | | |
| AS 3 | 0.846 | | | | | |
| AS 4 | 0.806 | | | | | |
| AS 5 | 0.814 | | | | | |
| DE 1 | | 0.742 | | | | |
| DE 2 | | 0.816 | | | | |
| DE 3 | | 0.793 | | | | |
| DE 4 | | 0.826 | | | | |
| DE 5 | | 0.780 | | | | |
| DE 6 | | 0.731 | | | | |
| DE 7 | | 0.684 | | | | |
| ILE 1 | | | 0.873 | | | |
| ILE 2 | | | 0.897 | | | |
| ILE 3 | | | 0.897 | | | |
| ILE 4 | | | 0.913 | | | |
| ILE 5 | | | 0.885 | | | |
| IS 1 | | | | 0.881 | | |
| IS 2 | | | | 0.894 | | |
| IS 3 | | | | 0.891 | | |
| IS 4 | | | | 0.822 | | |
| IS 5 | | | | 0.843 | | |
| LP 1 | | | | | 0.817 | |
| LP 2 | | | | | 0.800 | |
| LP 3 | | | | | 0.851 | |
| LP 4 | | | | | 0.744 | |
| LP 5 | | | | | 0.848 | |
| PD 1 | | | | | | 0.705 |
| PD 2 | | | | | | 0.761 |
| PD 3 | | | | | | 0.841 |
| PD 4 | | | | | | 0.862 |
| PD 5 | | | | | | 0.841 |
| PD 6 | | | | | | 0.822 |

**Table 2. Construction of reliability and validity of factors.**

| | Cronbach α | $\rho_a$ | $\rho_c$ | Average Variance Extracted |
|---|---|---|---|---|
| Acculturation Stress | 0.909 | 0.968 | 0.930 | 0.728 |
| Diversity Experience | 0.867 | 0.880 | 0.899 | 0.564 |
| Innovative Learning Environments | 0.937 | 0.938 | 0.952 | 0.798 |
| Institutional Support | 0.917 | 0.919 | 0.938 | 0.751 |
| Language Proficiency | 0.871 | 0.871 | 0.907 | 0.661 |
| Personal Development | 0.892 | 0.894 | 0.918 | 0.651 |

**Table 3. Various fitness indices obtained from a SEM investigation.**

|  | Saturated model | Estimated model |
|---|---|---|
| Standardized Root Mean Residual | 0.058 | 0.062 |
| d_ULS | 1.876 | 2.190 |
| d_G | 0.677 | 0.696 |
| Chi-square | 1407.040 | 1418.113 |
| Normed Fit Index | 0.849 | 0.848 |

**Table 4. Fornell-Larcker criterion.**

| Variables | AS | DE | ILE | IS | LP | PD |
|---|---|---|---|---|---|---|
| Acculturation Stress | 0.853 |  |  |  |  |  |
| Diversity Experience | 0.238 | 0.751 |  |  |  |  |
| Innovative Learning Environment | 0.260 | 0.664 | 0.893 |  |  |  |
| Institutional Support | 0.316 | 0.640 | 0.778 | 0.867 |  |  |
| Language Proficiency | 0.282 | 0.685 | 0.603 | 0.620 | 0.813 |  |
| Personal Development | 0.253 | 0.735 | 0.672 | 0.730 | 0.705 | 0.807 |

**Table 5. Path coefficients analyses revealing relationships between variables.**

|  | β | STDEV | t-statistics | p-values |
|---|---|---|---|---|
| DE → PD | 0.267 | 0.064 | 4.165 | 0.000 |
| ILE → PD | 0.098 | 0.066 | 1.477 | 0.140 |
| DE → LP | 0.508 | 0.061 | 8.336 | 0.000 |
| DE → AS | 0.118 | 0.071 | 1.655 | 0.098 |
| ILE → LP | 0.266 | 0.068 | 3.918 | 0.000 |
| ILE → AS | 0.182 | 0.074 | 2.449 | 0.014 |
| LP → PD | 0.210 | 0.052 | 4.061 | 0.000 |
| AS → PD | -0.036 | 0.036 | 1.016 | 0.310 |

and language proficiency is highly statistically significant ($\beta = 0.508$, p = 0.000). With a beta coefficient ($\beta$) of 0.508, a 1-unit change in diversity experience corresponds to a 50.8% change in Language Proficiency, confirming hypothesis $H_3$. The relationship between diversity experience and acculturation Stress is not statistically significant ($\beta = 0.118$, p = 0.098). Hence, results not support hypothesis $H_4$. The relationship between innovative learning environment and language proficiency is statistically significant ($\beta = 0.266$, p = 0.000). The beta coefficient ($\beta$) of 0.266 suggests that a 1-unit change in the innovative learning environment is expected to bring about a 26.6% change in Language Proficiency, providing support for hypothesis $H_5$. The relationship between innovative learning environment and Acculturation Stress is highly statistically significant ($\beta = 0.182$, p = 0.014). This implies that a 1-unit change in the innovative learning environment is associated with an 18.2% change in Acculturation Stress, supporting hypothesis $H_6$. The relationship between language proficiency and personal development is statistically significant ($\beta = -0.210, p = 0.000$), therefore, hypothesis $H_7$ is supported. The results show that the relationship between acculturation stress and personal development is not statistically significant ($\beta = -0.2036$, p = 0.310). While the coefficient indicates a negative relationship, the p-value is above the conventional 0.05 threshold, meaning this effect cannot be confidently confirmed. As a result, hypothesis H8, which proposed that acculturation stress negatively affects personal development, is not supported by the data. This

suggests that in this study, acculturation stress does not have a notable impact on students' personal development, and other factors may be more influential.

In Table 6 the specific indirect effect of diversity experience on personal development through language proficiency is 0.107. The T-statistic is 3.689, indicating that the relationship is highly statistically significant (p = 0.000). This suggests that there is a significant indirect effect of diversity experience on personal development through language proficiency, hence $H_9$ is approved. Furthermore, the specific indirect effect of an innovative learning environment on personal development through language proficiency is 0.056. The T-statistic is 2.782, indicating that the relationship is statistically significant (p = 0.005).

This suggests that there is a significant indirect effect of an innovative learning environment on personal development through language proficiency, thus $H_{10}$ is approved here. Moreover, the specific indirect effect of diversity experience on personal development through acculturation stress is -0.004. The T-statistic is 0.753, indicating that the relationship is not statistically significant (p = 0.452). This suggests that the indirect effect of diversity experience on personal development through acculturation stress is not significant in this analysis, so $H_{11}$ is not approved. Lastly, the specific indirect effect of an innovative learning environment on personal development through acculturation stress is -0.007.

The T-statistic is 0.882, which indicates that the relationship is not statistically significant (p = 0.378). This suggests that the indirect effect of an innovative learning environment on personal development through acculturation stress is not significant in this analysis, hence $H_{12}$ is also not approved. Fig 2 explains that institutional support strengthens the relationship between the diversity experience and personal development, so $H_{13}$ is approved. However, the analysis results do not show any moderation effect of institutional support on the relationship between innovative learning environments and personal development. Therefore,

**Table 6. Specific indirect effects.**

| Variable Relationship | β | Standard deviation | t-statistics | p-values |
|---|---|---|---|---|
| DE → LP → PD | 0.107 | 0.029 | 3.689 | 0.000 |
| ILE → LP → PD | 0.056 | 0.020 | 2.782 | 0.005 |
| DE → AS → PD | -0.004 | 0.006 | 0.753 | 0.452 |
| ILE → AS → PD | -0.007 | 0.008 | 0.882 | 0.378 |

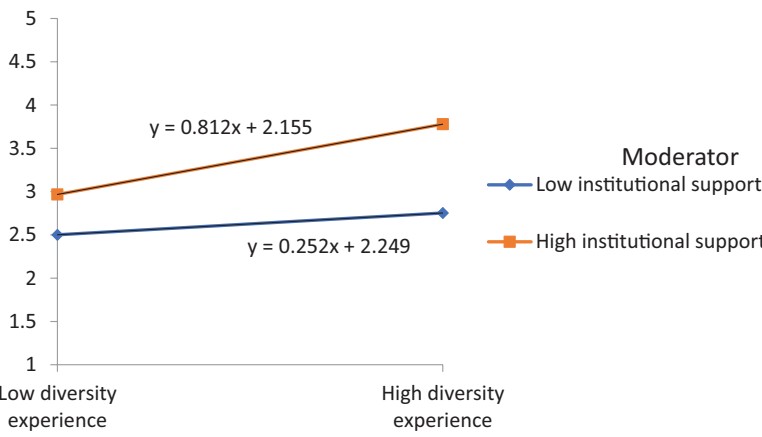

**Fig 2. The moderating effect on personal development.**

$H_{14}$ is rejected based on the information provided. Similarly, results also do not show an y moderation effect of institutional support on the relationship between language proficiency and personal development. Consequently, $H_{15}$ is rejected. Lastly, results also do not show any moderation effect of institutional support on the relationship between acculturation stress and personal development. Hence $H_{16}$ is rejected. Furthermore, the bootstrapping analysis depicted in Fig 3 explores the intricate relationships among diversity experiences, innovative learning environments, language proficiency, acculturation stress, institutional support, and personal development. Each construct is modeled as a latent variable, with multiple observed indicators exhibiting strong and significant loadings, thereby confirming their reliability as indicators of the respective constructs.

## 5. Discussion

This study sheds light on crucial factors affecting the personal development of international students in Shaanxi, China, including diversity experiences, innovative learning, language skills, acculturation stress, and institutional support. The findings can guide HEIs in enhancing student adaptation, satisfaction, and growth, ultimately fostering more supportive and effective environments for international learners. The results of this study demonstrate a statistically significant positive correlation between diversity experiences and personal development, suggesting that exposure to diverse cultural and social environments enhances students' personal growth. This finding aligns with previous research, which has shown that engaging with diverse perspectives promotes critical thinking, intercultural competence, and overall

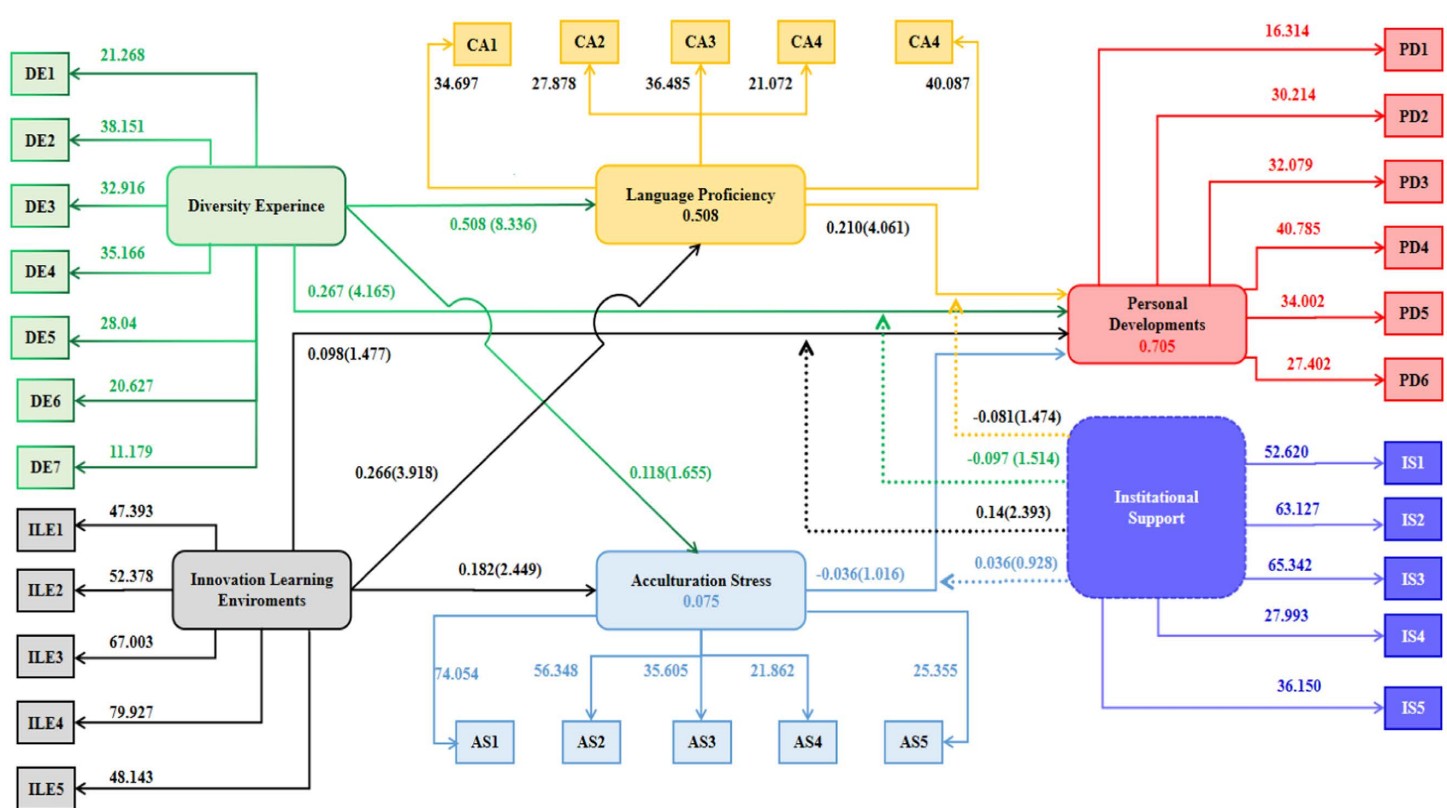

**Fig 3. The bootstrapping analysis of the variables of the study.**

personal development [41,47]. The capability to relate with individuals from different backgrounds lets international students to widen their global views and grow crucial skills such as compassion, intercultural sensitivity, and flexibility, all of which contribute significantly to their personal development.

The investigation revealed an insignificant correlation between innovative learning environments and personal development. A potential clarification for this result is that the impact of innovative learning environments may be secondary or mediated by other influences. Preceding study has highlighted that though innovative learning settings can stimulate engagement, peer learning, and creativity, their impact on personal development is often dependent on numerous individual aspects, such as student motivation, language proficiency, and the level of institutional support available. According to Boersma et al. [48] that while these environments boost academic collaboration, their effect on personal development is often mediated by the degree of student participation and the usefulness of feedback from educational bodies. Similarly, Yulianti and Badruzsaufari [49] indicated that critical thinking and personal development in such environments, substantial institutional support is crucial, embedding empowering instructional approaches and targeted measures.

Considering these outcomes, the results of this investigation imply that even though innovative learning environments can encourage academic exchange, their direct impact on personal development may remain constrained without institutional initiatives are designed to aid students. This could explain the insignificant connection identified in our findings, suggesting that the complete potential of innovative learning environments for nurturing personal growth might only be fully recognized, when shaped by influences like institutional support, active student engagement, or language competence.

The analysis uncovers a significant positive link between diversity experiences and language proficiency, showing that experience to diverse cultural environments markedly enhances students' proficiency in language skills. This finding is aligns with established literature, suggesting that interacting with people from diverse linguistic backgrounds offers students with chances to practice and improve their language proficiency in actual scenarios [50,51]. These interactions not only facilitate students become more fluent in the target language but also cultivating greater cultural insight and communication skills, each of which are integral to personal and academic growth.

The outcomes of the study specify a statistically non-significant association between diversity experience and acculturation stress, showing that simply engaging in varied cultural context may not be fully alleviate the psychological struggles faced by foreign students, during acculturation process. While earlier research has commonly shown that heighten interaction with varied groups, can reduce acculturation stress by enhancing social integration and intercultural competence [52,53], the inconclusive results in this study suggest that additional factors may be affecting the results. For example, aspect such as language proficiency, social care networks, and individual surviving strategies could have a more direct impact on dipping acculturative stress. This result suggests that diversity experiences, while valuable, may not be independently mitigate acculturative stress, and their effect possibly mediated by contributing factors.

The result show noteworthy correlations between innovative learning environments and language proficiency, in addition to significant correlations between innovative learning environments and acculturation stress. These findings propose that participation in innovative educational settings may support language skills while also influencing how students experience and manage acculturation stress. Prior studies have shown that technologically advanced and interactive academic environments not only provision language development by providing real-world communication opportunities but also aid reduce the psychological strain of acculturation by nurturing social engagement and peer collaboration [19,54].

The link between language proficiency and personal development is statistically considerable, representing that advanced language abilities offer valuable input to the personal development of foreign students. This outcomes is in agreement with earlier studies, that have highlighted the critical importance of language proficiency in strengthening the self-confidence, intercultural adaptability, and social belonging, each of which are indispensable for personal development [55,56]. fluency in the local language permits students to handle both academic and social contexts more confidently, thereby fostering their complete personal development.

The link between acculturation stress and personal development in this research observed to be non-significant, showing that the psychological encounters of acculturation may not have a direct effect on students' personal development. This result varies from prior research that commonly point out the negative consequences of acculturation stress on personal development, predominantly for international students [57,58]. One reason for this outcome could be the existence of institutional support. Studies have exposed that when educational education afford resources such as counseling, peer mentoring, and social interaction network, they aid ease the negative influence of acculturation stress, possibly weakening its direct impact on personal development [59]. In this situation, the existence of support mechanisms may have decreased the influence of acculturation stress, revealing the insignificant association detected in this research.

Another likely clarification is the key role of specific handling approaches and resilience. Students who advance active coping strategy, such as looking for social assistance or engaging in campus events, are generally better able to handle acculturative hurdles, letting them to manage personal development in spite of the existence of stress [60]. Also, language proficiency can also contribute an important role. Students who demonstrate superior language proficiency often deal with fewer hindrances to social and learning integration, which can alleviate the effects of acculturation stress on personal development [56]. Although acculturation stress may be a major component of student experience, its influence on personal development may be mediated by other factors, such as institutional support, coping mechanisms, and language skills.

The findings of the study show a statistically significant indirect effect of diversity experience on personal development through language proficiency, confirming that though diversity experiences alone may not directly influence personal development, their influence is mediated by improved language skills. This is consistent with prior study that showing language proficiency is vital for enabling cross-cultural capability and social interaction, both of which contribute to personal development [56]. Additionally, introduction to diverse environments provides students with opportunities to practice and improve their language skills, which supports their complete adaptation and personal development [51]. These outcomes reveal that nurturing diversity in educational settings, together with contribution language assistance, can significantly improve personal development of international students. Furthermore, the results reveal a statistically significant indirect effect of innovative learning environments on personal development through language proficiency, showing that these environments foster personal development by improving language proficiency. This result in line with present result, which proposes that innovative learning environments endorse language acquisition by creating immersive and interactive learning opportunities, in this way supporting students' enhance learning and personal development [48]. Additionally, language proficiency has been shown to mediate the relationship between learning environments and personal development, as better language proficiency enable students to interact extra effectively in schooling and social contexts, further contributing to their personal development [49].

The findings discloses that the indirect influence of both diversity experience and innovative learning environments on personal development through acculturation stress are not statistically significant. This advocates that acculturation stress does not exert an impactful mediating role. Former has commonly pointed out that the negative effect of acculturation stress on students' personal growth [52,57], but in context of this study, the outcome show that additional factors may be at play. One explanation could be the existence of institutional support network, such as peer mentoring and counseling, which help bumper the negative impact of acculturation stress [19]. Additionally, language proficiency and social support networks may provide a more direct route to personal development, dipping the impact of acculturation stress as a [56]. These result show that further study can explore the role of these moderating factors in the association between diversity experiences, learning environments, and student personal development.

The result reveal that institutional support act as a critical moderating in the association between diversity experience and personal development. While diversity experiences alone potentially lacking direct influence personal development, the presence institutional support, such as language assistance, learning advising, and peer mentoring, can significantly improve the influence of these experiences. Past study reveals that student who interact with diverse cultural environments are optimally placed for personal development, when they have access to resources that enable their integration and academic capabilities [19]. Institutional support advances the link between diversity experiences and personal development by serving students navigate the challenges of adapting to new social and academic contexts, ultimately indorsing their complete development [52]. This shows that institutions must not only promote diversity but also offer structured support systems that permit students to fully advantage from these experiences.

Moreover, together SCT and EST offer an inclusive outline for understanding the intricate factors impacting foreign students' personal development. SCT highlights the role of self-efficacy and social learning, while EST positions students within a wider network of support systems, revealing the complex part of institutional structures. Even though, these theories offer a strong building for understanding foreign students' practices, the results suggest potential areas for theoretical growth. For example, joining resilience-focused backing within the exosystem could discourse the exceptional encounters by international students, enhancing their capability to familiarize and thrive in diverse academic environments. These understandings underscore the transformative potential of well-supported educational environments, where students are allowed to steer challenges and realize their full potential. Future study may help to exploring additional theoretical frameworks or cultural dimensions to further capture the exclusive experiences of foreign students.

## 6. Conclusions and policy recommendations

This study examined how diversity experiences, innovative learning environments, language proficiency, acculturation stress, and institutional support influence the personal development of international students in HEIs in Shaanxi, China. The results underline the serious role of diversity experiences and language proficiency in promoting personal development, representing that cross-cultural connections and effective communication skills are instrumental in structure confidence, flexibility, and self-efficacy among foreign students. Additionally, while innovative learning environments did not show a direct influence on personal development, institutional support arose as a key element in reducing acculturation stress, underlining the importance of supportive academic building in improving students' academic and personal development.

These findings underscore the essential for academic institutions to nurture diversity experiences and strengthen support systems for foreign students. Academic institutions can

improve students' engagement and reduce acculturation stress through initiatives like mentorship programs pairing international and local students, intercultural workshops, and tailored language support services. Creating inclusive learning environments that revel diversity not only assist foreign students adapt and thrive but also nurtures a sense of belonging, encouraging them to fully engage in their academic and learning journeys.

Despite these significant understandings, the research has certain limitations. first the sample was restricted to international students in HEIs within Shaanxi, China, which may limit the generalizability of the findings to students in other areas or academic systems with changing cultural dynamics. Second, the study depended entirely on quantitative survey data, which may not fully capture the depth of students' experiences, predominantly in relation to acculturation and resilience. Furthermore, future study can discourse these limitations by growing the sample to comprise a wider geographic range and leading comparative analyses of international students' experiences across diverse cultural contexts. Incorporating qualitative approaches, such as interviews or focus groups, could also offer valuable understanding into the emotional and psychological sides of student adaptation. Further investigation might also discover resilience-focused support initiatives within HEIs to govern how targeted programs improve foreign students' capability to flourish and develop in different academic context.

## Supporting information

**S1 File. Research data for this study.**
(DOCX)

**S2 File. Research data for this study.**
(XLSX)

**S3 File. Research data for this study.**
(XLSX)

## Author contributions

**Conceptualization:** Muhammad Azram, Mei Hong, Waqar Ahmad.

**Data curation:** Ali Sohail.

**Funding acquisition:** Mei Hong.

**Investigation:** Muhammad Azram, Ali Sohail.

**Methodology:** Muhammad Azram, Waqar Ahmad, Ali Sohail.

**Project administration:** Muhammad Azram, Mei Hong.

**Software:** Waqar Ahmad.

**Supervision:** Mei Hong.

**Validation:** Waqar Ahmad.

**Visualization:** Mei Hong.

**Writing – original draft:** Muhammad Azram.

**Writing – review & editing:** Muhammad Azram, Ali Sohail.

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
