## [Decision Letter · Decision Letter 0]

2 Oct 2024

PONE-D-24-35389Exploring the Relationship of Diversity Experiences, Innovative Learning Environments, and Personal Development among International Students in ChinaPLOS ONE

Dear Dr. Azram,

Thank you for submitting your manuscript to PLOS ONE. After careful consideration, we feel that it has merit but does not fully meet PLOS ONE’s publication criteria as it currently stands. Therefore, we invite you to submit a revised version of the manuscript that addresses the points raised during the review process.

#Academic Editor Comment:  "Given the reviewers' feedback, major revisions are required to address the identified shortcomings and improve the overall quality of the manuscript."

We look forward to receiving your revised manuscript.

Kind regards,

Abeer F. Alkhwaldi

Academic Editor

PLOS ONE

Journal Requirements: When submitting your revision, we need you to address these additional requirements. 1. Please ensure that your manuscript meets PLOS ONE's style requirements, including those for file naming. The PLOS ONE style templates can be found at https://journals.plos.org/plosone/s/file?id=wjVg/PLOSOne_formatting_sample_main_body.pdf and https://journals.plos.org/plosone/s/file?id=ba62/PLOSOne_formatting_sample_title_authors_affiliations.pdf 2. Please include your full ethics statement in the ‘Methods’ section of your manuscript file. In your statement, please include the full name of the IRB or ethics committee who approved or waived your study, as well as whether or not you obtained informed written or verbal consent. If consent was waived for your study, please include this information in your statement as well. 3. Please ensure that you refer to Figure 1 and 2 in your text as, if accepted, production will need this reference to link the reader to the figure. 4. We note you have included a table to which you do not refer in the text of your manuscript. Please ensure that you refer to Table 1 in your text; if accepted, production will need this reference to link the reader to the Table. 5. Please include captions for your Supporting Information files at the end of your manuscript, and update any in-text citations to match accordingly. Please see our Supporting Information guidelines for more information: http://journals.plos.org/plosone/s/supporting-information.

**Additional Editor Comments:**

Academic Editor Comment: "Given the reviewers' feedback, major revisions are necessary"

Reviewers' comments:

Reviewer's Responses to Questions

**Comments to the Author**

1. Is the manuscript technically sound, and do the data support the conclusions?

Reviewer #1: Yes

Reviewer #2: No

Reviewer #3: Yes

2. Has the statistical analysis been performed appropriately and rigorously? 

Reviewer #1: Yes

Reviewer #2: No

Reviewer #3: I Don't Know

3. Have the authors made all data underlying the findings in their manuscript fully available?

Reviewer #1: No

Reviewer #2: Yes

Reviewer #3: Yes

4. Is the manuscript presented in an intelligible fashion and written in standard English?

Reviewer #1: Yes

Reviewer #2: No

Reviewer #3: Yes

5. Review Comments to the Author

Reviewer #1: The research explores relevant and contemporary issues in higher education, focusing on the role of diversity experiences, innovative learning environments, and personal development among international students in China. Overall, the paper is technically sound and well-supported by the data, but there are areas where the manuscript could benefit from clarification or additional detail, particularly with regard to non-significant results and data availability. There are several suggestions for improvement;

- The relationships that were found to be non-significant (e.g., acculturation stress and personal development), a more in-depth discussion could help contextualize these results and explain why they may differ from expectations.

- The authors could provide more explanation of SEM assumptions and interpretations, particularly around the non-significant relationships. Additionally, explaining the practical significance or effect sizes of significant relationships would strengthen the conclusions.

-The authors should either provide anonymized data or a more detailed explanation of the restrictions, especially if privacy or ethical considerations are involved. If possible, providing access to the data through a repository or institutional database, with clear guidelines for researchers seeking access, would be ideal.

-There are a few minor typographical and grammatical issues that could be corrected during revision, but they do not detract significantly from the overall readability of the manuscript.

-The practical implications of the findings are discussed briefly but could be expanded to provide more concrete recommendations for educators and institutions, particularly with regard to fostering diversity experiences and providing institutional support for international students.

Reviewer #2: Abstract – It is recommended to explain the variables before presenting the findings, as readers may not understand if the author jumps directly to the results.

Literature Review – Do not confuse this section with the introduction. It is difficult to locate the research questions. The variables should be further elaborated on through precedent studies, including both positive and negative outcomes. This will help readers identify the research gaps.

Methodology – The section needs more elaboration, such as why these participants were chosen, how the sampling was obtained, and whether the variables were tested in a pilot study. Additionally, only the quantitative design is mentioned. The authors should provide more details about their approach in the quantitative study.

Findings – This section seems fine.

Discussion – The discussion is lacking. There is little mention of the relationship between the theoretical frameworks used and the findings. Moreover, the discussion lacks critical analysis.

Conclusion – The authors did not conclude the study effectively. There are no suggestions for future research or any mention of the study’s limitations.

Technicality – There are numerous grammatical and citation errors. The authors need to adhere to a consistent citation style. Additionally, capital letters are used excessively, even where not required (e.g., for non-special pronouns).

Reviewer #3: Dear authors,

Your paper is good and interesting. Yet, there are many issues that need to consider to enrich the paper and to make it publishable. There are some scientific and technical issues that need to be reconsidered. This is never meant to underestimate the work. The review is attached.

6. PLOS authors have the option to publish the peer review history of their article (what does this mean? ). If published, this will include your full peer review and any attached files.

**Do you want your identity to be public for this peer review?** For information about this choice, including consent withdrawal, please see our Privacy Policy .

Reviewer #1: **Yes: ** ZULHASNI BIN ABDUL RAHIM

Reviewer #2: No

Reviewer #3: **Yes: ** Nawal Fadhil Abbas

---

## [Author Response · Author response to Decision Letter 1]

2 Nov 2024

Response to the Editor

Journal Requirements

Comment 1: When submitting your revision, we need you to address these additional requirements.

Response 1: We have revised the manuscript to ensure it aligns with PLOS ONE's style requirements, as outlined in the provided templates. Specific file naming and formatting adjustments have been made as required.

Comment 2: Please include your full ethics statement in the ‘Methods’ section of your manuscript file. In your statement, please include the full name of the IRB or ethics committee who approved or waived your study, as well as whether or not you obtained informed written or verbal consent. If consent was waived for your study, please include this information in your statement as well.

Response 2: A full ethics statement has been added to the Methods section, including the name of the Institutional Review Board (IRB) that approved the study, details on informed consent, and a note on whether consent was waived.

Comment 3. Please ensure that you refer to Figure 1 and 2 in your text as, if accepted, production will need this reference to link the reader to the figure.

Response 3: We have ensured that Figures 1 and 2 are referred to in the manuscript text to facilitate production and reader navigation.

Comment 4. We note you have included a table to which you do not refer in the text of your manuscript. Please ensure that you refer to Table 1 in your text; if accepted, production will need this reference to link the reader to the Table.

Response 4: Table 1 is now explicitly referenced in the text to guide readers and align with production requirements.

Comment 5. Please include captions for your Supporting Information files at the end of your manuscript, and update any in-text citations to match accordingly. Please see our Supporting Information guidelines for more information: http://journals.plos.org/plosone/s/supporting-information

Response 5: Captions for all Supporting Information files have been included at the end of the manuscript, and in-text citations have been updated accordingly to comply with PLOS ONE’s guidelines.

Response to Reviewers

Thank you for your insightful comments and suggestions. We have carefully addressed each point to improve the manuscript. Below are our responses:

Reviewer #1

The research explores relevant and contemporary issues in higher education, focusing on the role of diversity experiences, innovative learning environments, and personal development among international students in China. Overall, the paper is technically sound and well-supported by the data, but there are areas where the manuscript could benefit from clarification or additional detail, particularly with regard to non-significant results and data availability. There are several suggestions for improvement;

Comment 1: The relationships that were found to be non-significant (e.g., acculturation stress and personal development), a more in-depth discussion could help contextualize these results and explain why they may differ from expectations.

Response 1: Thank you for your suggestion. We have expanded the discussion on non-significant relationships, specifically between acculturation stress and personal development. This addition provides context for these results and explores potential reasons for the lack of significance.

Please see line from 437 to 455, 464 to 474, 492 to 512, and 533 to 545. They are highlighted in yellow color.

Comment 2: The authors could provide more explanation of SEM assumptions and interpretations, particularly around the non-significant relationships. Additionally, explaining the practical significance or effect sizes of significant relationships would strengthen the conclusions.

Response 2: Additional details have been provided regarding the assumptions underlying Structural Equation Modeling (SEM) and interpretations of non-significant relationships. We have also discussed the effect sizes of significant relationships to clarify their practical significance.

Please see line from 327 to 332. which are highlighted in yellow color.

Comment 3: The authors should either provide anonymized data or a more detailed explanation of the restrictions, especially if privacy or ethical considerations are involved. If possible, providing access to the data through a repository or institutional database, with clear guidelines for researchers seeking access, would be ideal.

Response 3: We have updated the data availability statement, clarifying the availability of anonymized data and any restrictions due to ethical considerations. An institutional repository has been suggested for access, with clear guidelines for researchers seeking permission.

Please see line from 334 to 339 which are highlighted in yellow color.

Comment 4: There are a few minor typographical and grammatical issues that could be corrected during revision, but they do not detract significantly from the overall readability of the manuscript.

Response 4: We have reviewed the manuscript for typographical and grammatical errors, making corrections where necessary to enhance readability.

Comment 5: The practical implications of the findings are discussed briefly but could be expanded to provide more concrete recommendations for educators and institutions, particularly with regard to fostering diversity experiences and providing institutional support for international students.

Response 5: The Discussion section has been expanded to offer more concrete recommendations for educators and institutions on fostering diversity experiences and supporting international students. These enhancements aim to provide actionable insights based on our findings.

Please see line from 586 to 592. Which are highlighted in yellow color.

REVIEWER #2

We appreciate your detailed feedback, which has been instrumental in refining the manuscript. Please find our responses below:

Comment 1:

Abstract: It is recommended to explain the variables before presenting the findings, as readers may not understand if the author jumps directly to the results.

Response 1: We have revised the Abstract to clarify the variables before presenting the findings, ensuring that readers can understand the context of the results.

Please see the “Abstract’, which is highlighted in yellow color.

Comment 2:

Literature Review – Do not confuse this section with the introduction. It is difficult to locate the research questions. The variables should be further elaborated on through precedent studies, including both positive and negative outcomes. This will help readers identify the research gaps.

Response 2: We have reorganized the Literature Review to distinguish it more clearly from the Introduction, with a focus on explaining variables and discussing prior studies. The research questions are now stated explicitly to improve clarity.

Please see line from 3 to 239. Which are highlighted in yellow color.

Comment 3:

Methodology – The section needs more elaboration, such as why these participants were chosen, how the sampling was obtained, and whether the variables were tested in a pilot study. Additionally, only the quantitative design is mentioned. The authors should provide more details about their approach in the quantitative study.

Response 3: Additional details have been included in the Methodology section, explaining the rationale for participant selection, sampling approach, and the quantitative design. We have also specified whether a pilot study was conducted.

Please see line from 280 to 339. Which are highlighted in yellow color.

Comment 4:

Findings – This section seems fine.

Response 4: Thank you very much.

Comment 5:

Discussion – The discussion is lacking. There is little mention of the relationship between the theoretical frameworks used and the findings. Moreover, the discussion lacks critical analysis.

Response 5: The Discussion has been enhanced to deepen the connection between the theoretical frameworks (Social Cognitive Theory and Ecological Systems Theory) and the findings, along with a more critical analysis of the study’s implications.

Please see line from 428 to 574. Which are highlighted in yellow color.

Comment 6:

Conclusion – The authors did not conclude the study effectively. There are no suggestions for future research or any mention of the study’s limitations.

Response 6: The Conclusion section has been revised to include specific suggestions for future research and a discussion of the study’s limitations, addressing your feedback directly. Please see line from 576 to 604. Which are highlighted in yellow color.

Comment 7:

Technicality – There are numerous grammatical and citation errors. The authors need to adhere to a consistent citation style. Additionally, capital letters are used excessively, even where not required (e.g., for non-special pronouns).

Response 7: We have ensured consistency in citation style throughout the manuscript, corrected excessive use of capital letters, and improved grammar and readability by addressing common errors.

REVIEWER #3

Thank you for your constructive comments. Below are our responses, detailing the revisions made based on your suggestions:

Comment 1: The first finding stated in the abstract reveals the existence of a positive association between experience and personal development while the first sentence in the abstract focuses on the association between experience and innovative learning environment and their impact on personal development. Do the two sentences mean the same? If yes, please re-consider the introductory statement of the abstract.

Response 1: Thank you for your suggestion. We have revised the introductory statement in the Abstract to clarify the distinction between diversity experiences and innovative learning environments, ensuring that each is presented with a clear, distinct impact on personal development. Please see the revised abstract.

Comment 2: Some recommendations or pedagogical implications are preferable to end up the abstract with.

Response 2: We have added a brief mention of pedagogical implications at the end of the Abstract, providing concrete applications of the study’s findings for educators and institutions. Please see the revised abstract.

Comment 3: The research gap and significance are not clearly stated in the abstract.

Response 3: We have updated the Abstract to clearly state the research gap and the significance of this study, emphasizing the contributions of our work to the field. Please see the revised abstract.

Comment 4: Referencing the source in “… are emphasized by Souto-Manning, Falk et al. (2019).” is not proper. Only the last name of the first author is used followed by ‘et al’. Follow the APA system in this regard, the same with the other sources in the second and other paragraphs of the Introduction section.

Response 4: We have revised the in-text citations to ensure they adhere to APA guidelines, using “et al.” only after the first author’s name. All similar issues throughout the manuscript have been corrected. Please see the manuscript thoroughly.

Comment 5: In the in-text citation, the ampersand ‘&’ is used to connect two names, while ‘and’ is used in the text only. Correct accordingly.

Response 5: We have corrected all in-text citations to ensure proper use of “&” in parenthetical citations and “and” in narrative citations, in line with APA style.

Comment 6: Conclusion – The authors did not conclude the study effectively. There are no suggestions for future research or any mention of the study’s limitations.

Response 6: The Conclusion section has been revised to include specific suggestions for future research and a discussion of the study’s limitations, directly addressing your feedback. Please see the revised conclusion, which are highlighted in yellow.

Comment 7: Why do abbreviations such as PD, ILE, and DE appear in the Hypotheses section? They should appear the first time the words are used in the Introduction section.

Response 7: We have now introduced the abbreviations “PD,” “ILE,” and “DE” in the Introduction section when the terms are first mentioned, ensuring clarity for readers.

Comment 8: Editing is highly recommended to fix such language issues as in “… the relationship between diversity experiences and personal development which hypnotized as H1 is exist according to the literature.” and other places.

Response 8: We have thoroughly reviewed the manuscript to address grammatical issues and awkward phrasing, including the specific example provided. Additional edits have been made to improve readability.

Comment 9: A paragraph should start with the authors’ names as a subject followed by a year between round brackets. Do not start a sentence with a round bracket as in “(Katsiaficas, Suárez-Orozco, et al. 2013) suggested that acculturation stress can play a mediating …”.

Response 9: We have revised instances where sentences began with parentheses, placing authors’ names at the beginning of sentences and following with the publication year in parentheses, ensuring improved sentence flow.

Comment 10: The hypotheses section is very long and confusing. I suggest grouping the hypotheses after reducing them to 5-6 and under the title ‘Hypotheses’ and grouping the studies under the title ‘Previous Studies’.

Response 10: The Hypotheses section has been restructured to improve clarity. Hypotheses are now grouped under the title “Hypotheses,” and supporting studies are summarized under a new section titled “Previous Studies.”

Comment 11: I suggest having a note below Table 1 referring to the abbreviations included in the table to make it easier for the reader to understand the study variables.

Response 11: A note has been added below Table 1, defining all abbreviations used. This addition enhances readability by helping readers interpret the study variables.

Comment 12: After the Introduction and before the Results and Discussions, the researchers need to have two main sections: Literature Review and Methodology.

Response 12: The manuscript has been reorganized to include separate sections for the Literature Review and Methodology, positioned between the Introduction and Results and Discussion sections.

Comment 13: Under the section Results and Discussion, there should be two sub-sections: Results and Discussion.

Response 13: The Results and Discussion section is now divided into two distinct subsections, “Results” and “Discussion,” separately, improving clarity and separation of findings from interpretations.

Comment 14: I suggest having the same point raised in 11 above for Table 2.

Response 14: A similar note has been added below Table 2 to explain abbreviations, following the approach used for Table 1.

Comment 15: The researchers need to end their paper with recommendations or pedagogical implications.

Response 15: The Conclusion section has been revised to include specific recommendations and pedagogical implications based on our findings, offering practical guidance for educators and institutions.

Comment 16: The whole reference list should be re-considered in light of the APA system of documentation.

Response 16: We have reviewed and reformatted the entire reference list to ensure compliance with APA style, correcting any inconsistencies and formatting errors.

---

## [Decision Letter · Decision Letter 1]

20 Nov 2024

The Influence of Language Proficiency, Acculturation Stress, and Institutional Support in enhancing Personal Development of International Students in China

PONE-D-24-35389R1

Dear Dr. Azram,

We’re pleased to inform you that your manuscript has been judged scientifically suitable for publication and will be formally accepted for publication once it meets all outstanding technical requirements.

Kind regards,

Xiaoguang Fan

Academic Editor

PLOS ONE

Additional Editor Comments (optional):

Reviewers' comments:

Reviewer's Responses to Questions

**Comments to the Author**

1. If the authors have adequately addressed your comments raised in a previous round of review and you feel that this manuscript is now acceptable for publication, you may indicate that here to bypass the “Comments to the Author” section, enter your conflict of interest statement in the “Confidential to Editor” section, and submit your "Accept" recommendation.

Reviewer #2: All comments have been addressed

Reviewer #3: All comments have been addressed

2. Is the manuscript technically sound, and do the data support the conclusions?

Reviewer #2: Yes

Reviewer #3: Yes

3. Has the statistical analysis been performed appropriately and rigorously? 

Reviewer #2: Yes

Reviewer #3: I Don't Know

4. Have the authors made all data underlying the findings in their manuscript fully available?

Reviewer #2: Yes

Reviewer #3: Yes

5. Is the manuscript presented in an intelligible fashion and written in standard English?

Reviewer #2: Yes

Reviewer #3: Yes

6. Review Comments to the Author

Reviewer #2: There are still few errors such as in the title. "in enhancing" should replace with "in Enhancing". Certain authors' name are written in capital letters such as MEI et al. (2018). In terms of analysis part, there are too many hypotheses in one paper. While I believe that they did well in their findings, nevertheless, the readability of this paper is questionable.

Reviewer #3: Dear author,

Thank you for addressing most of the comments. Yet, very few issues are still to be covered:

- There is no note below table 1 to define the abbreviated forms.

- Figure 1 should be placed after line 277, not between lines 276 and 277.

- For articles from journals, not all of them are followed by the range of pages, i.e., from...to.

7. PLOS authors have the option to publish the peer review history of their article (what does this mean? ). If published, this will include your full peer review and any attached files.

**Do you want your identity to be public for this peer review?** For information about this choice, including consent withdrawal, please see our Privacy Policy .

Reviewer #2: **Yes: ** KAMARUL AZAM

Reviewer #3: **Yes: ** Nawal Fadhil Abbas

---

## [Editor Report · Acceptance letter]

PONE-D-24-35389R1

PLOS ONE

Dear Dr. Azram,

I'm pleased to inform you that your manuscript has been deemed suitable for publication in PLOS ONE. Congratulations! Your manuscript is now being handed over to our production team.

Kind regards,

on behalf of

Dr. Xiaoguang Fan

Academic Editor

PLOS ONE